# Detection of *TEM* and *CTX-M* Genes in *Escherichia coli* Isolated from Clinical Specimens at Tertiary Care Heart Hospital, Kathmandu, Nepal

**DOI:** 10.3390/diseases9010015

**Published:** 2021-02-07

**Authors:** Ram Shankar Prasad Sah, Binod Dhungel, Binod Kumar Yadav, Nabaraj Adhikari, Upendra Thapa Shrestha, Binod Lekhak, Megha Raj Banjara, Bipin Adhikari, Prakash Ghimire, Komal Raj Rijal

**Affiliations:** 1Central Department of Microbiology, Tribhuvan University, Kirtipur, Kathmandu 44618, Nepal; shankarram224@gmail.com (R.S.P.S.); bwith.binod@gmail.com (B.D.); adhikarinaba13@gmail.com (N.A.); upendrats@gmail.com (U.T.S.); binodlekhak9@gmail.com (B.L.); banjaramr@gmail.com (M.R.B.); prakashghimire@gmail.com (P.G.); 2Shahid Gangalal National Heart Centre, Bansbari, Kathmandu 44618, Nepal; yadavbinod4u@gmail.com; 3Mahidol Oxford Tropical Medicine Research Unit, Faculty of Tropical Medicine, Mahidol University, Bangkok 10400, Thailand; biopion@gmail.com

**Keywords:** uropathogenic *E. coli*, ESBL, antimicrobial resistance, Temoneira (TEM), Cefotaximase, *bla*_CTX-M_, *bla*_TEM_, Nepal

## Abstract

Background: Antimicrobial resistance (AMR) among Gram-negative pathogens, predominantly ESBL-producing clinical isolates, are increasing worldwide. The main aim of this study was to determine the prevalence of ESBL-producing clinical isolates, their antibiogram, and the frequency of ESBL genes (*bla*_TEM_ and *bla*_CTX-M_) in the clinical samples from patients. Methods: A total of 1065 clinical specimens from patients suspected of heart infections were collected between February and August 2019. Bacterial isolates were identified on colony morphology and biochemical properties. Thus, obtained clinical isolates were screened for antimicrobial susceptibility testing (AST) using modified Kirby–Bauer disk diffusion method, while ESBL producers were identified by using a combination disk diffusion method. ESBL positive isolates were further assessed using conventional polymerase chain reaction (PCR) to detect the ESBL genes *bla*_TEM_ and *bla*_CTX-M_. Results: Out of 1065 clinical specimens, 17.8% (190/1065) showed bacterial growth. Among 190 bacterial isolates, 57.4% (109/190) were Gram-negative bacteria. Among 109 Gram-negative bacteria, 40.3% (44/109) were *E. coli*, and 30.2% (33/109) were *K. pneumoniae*. In AST, 57.7% (*n* = 63) Gram-negative bacterial isolates were resistant to ampicillin and 47.7% (*n* = 52) were resistant to nalidixic acid. Over half of the isolates (51.3%; 56/109) were multidrug resistant (MDR). Of 44 *E. coli*, 27.3% (12/44) were ESBL producers. Among ESBL producer *E. coli* isolates, 58.4% (7/12) tested positive for the *bla*_CTX-M_ gene and 41.6% (5/12) tested positive for the *bla*_TEM_ gene. Conclusion: Half of the Gram-negative bacteria in our study were MDR. Routine identification of an infectious agent followed by AST is critical to optimize the treatment and prevent antimicrobial resistance.

## 1. Background

*Escherichia coli* and *Klebsiella* species comprise the largest portion of the Gram-negative pathogens in several nosocomial and community-acquired infections, such as intra-abdominal infection, bloodstream infection (BSI), meningitis, and pyogenic liver abscess (PLA) [1]. *E. coli* is a commensal of the mammalian gastrointestinal tract, but can also be found in water, soil, and vegetation [2]. Virtually all pathogenic strains of Gram-negative bacteria are responsible for several infections, including gastroenteritis, urinary tract infection (UTI), septicemia, nosocomial infections, pneumonia, brain and abdominal abscess, and neonatal meningitis [3].

Antimicrobials are the most effective choice of drugs against infectious bacteria. Fluroquinolones, cephalosporins, β-lactams, and β-lactamases inhibitors alone or in combination are the frequently prescribed drugs in response to infections caused by Gram-negative bacteria [4]. Nonetheless, overuse of such drugs in humans, animals, and the environment is deemed responsible for the emergence of antibiotic resistance [5]. Antimicrobial resistance (AMR) is the condition in which pathogenic strains of the bacteria develop resistance against the specific drug prescribed in response to that microorganism(s) [6,7]. For instance, Gram-negative bacteria have developed resistance to one of the most effective drugs (β-lactams) by producing enzymes that can hydrolyze bonds of β-lactam rings. These enzymes are named as extended-spectrum β-lactamases (ESBLs), which possess the ability to inactivate extended-spectrum β-lactams and monobactams, except cephamycins and imipenem [8]. ESBLs are Ambler Class A β-lactamases that have different genotypic variants, such as *bla*_SHV_, *bla*_TEM_, and *bla*_CTX-M_ [8]. In addition to this, ESBL producers exhibit co-expression of many other genes. Other less frequently recovered variants are bla KPC, VEB, PER, BEL-1, BES-1, SFO-1, TLA, and bla BIC [8].

The emergence and spread of these drug-resistant genes can limit therapeutic options, increase morbidities and mortalities, prolong hospital stays, and cost massive economic loss [9]. Most of the low and middle income countries (LMICs), including Nepal, have poor infection control strategies, and lack of diagnostic facilities, routine monitoring, and surveillance systems for AMR [10,11,12]. In addition to this, wide availability and use of over-the-counter (OTC) antimicrobials are further aggravating the AMR in LMIC settings, including Nepal [13].

The prevalence of ESBL-producing pathogens varies widely, even in closely related regions. Higher rates of ESBLs have been reported from Southeast Asia, including Nepal [14]. Various studies have reported the circulation of ESBL-producing bacterial strains in Nepal [3,6,15,16,17,18]. Laboratory facilities for culture and antimicrobial susceptibility testing in different peripheral/district hospitals are still not available in Nepal. Inadequate laboratory facilities can result in the improper diagnosis of the disease, and thus lead to the use of inappropriate antibiotics [19]. Clinical suspicion of the ß-lactamases in the absence of proper diagnostic methods may have increased the rate of ESBLs in Nepal. It is critical to understand the dynamics of antimicrobial resistance within specific contexts for choosing effectiveness of life-saving antibiotics and to control ESBLs-producing pathogens [3,20].

Drug resistance is often difficult to recognize by using conventional antibiotic sensitivity test methods alone [5]. Failure to identify ESBL-producing organisms contributes to their uncontrolled spread [12]. Identification of the resistant phenotype is crucial in preventing the AMR, more so in developing countries, where there is excessive use of antibiotics and a lack of adequate antimicrobial resistance surveillance [11,17]. The main objective of this study was to explore the prevalence of ESBL-producing Gram-negative bacteria, their antibiogram, and molecular detection of plasmid-mediated ESBL genes *bla*_TEM_ and *bla*_CTX-M_ in *E. coli* among patients suspected of heart infections at Gangalal Heart Hospital, Kathmandu, Nepal. 

## 2. Material and Methods

### 2.1. Study Design, Study Site, and Sample Population

This hospital-based cross-sectional study was conducted at the microbiology laboratory of Shahid Gangalal National Heart Centre, Bansbari, Kathmandu and the Central Department of Microbiology, Tribhuvan University, Kathmandu, Nepal between February and August 2019. This study population was inclusive of all genders and age groups (from newborn to 90 years) attending the hospital (inpatient and outpatient department) with suspicion of heart infections. All of the study subjects provided written informed consent prior to enrolment in the study. 

### 2.2. Sample Size and Sample Type

A total of 1065 non-duplicated clinical specimens were collected from the patients attending Shahid Gangalal National Heart Center. Among 1065 clinical specimens analyzed, 246 were blood, 300 were urine, 280 were sputum, 151 were pus or wounds, 59 were tips (catheter tips, central venous pressure tips, endotracheal tube, and secretion and suction tips), and 25 were fluids (pericardial fluids, peritoneal fluids, pleural fluids, and other body fluids). Demographic information was collected using a structured questionnaire. 

### 2.3. Sample Collection and Transportation

All of the samples were collected in a clean, leak-proof container by the trained medical personnel. After aseptic collection of the samples, they were well-labeled and immediately sent to the microbiology department for further assay. In case of delay in delivery, samples were either refrigerated or preserved at 4 °C with appropriate preservatives. Samples were collected and processed by following standard microbiological methods [21,22]. 

### 2.4. Laboratory Processing of the Specimens 

#### 2.4.1. For Blood Samples

Five ml of blood were mixed with 45 mL of brain heart infusion (BHI) broth for adults and 1 mL of blood with 9 mL of BHI broth for children. The bottle was incubated at 37 °C for seven days. After sufficient incubation, bottles showing turbidity were further sub-cultured aerobically in MacConkey agar (MA) and blood agar (BA) at 37 °C for 24 to 48 h. If growth occurred, then the isolated colony was identified by colony morphology, Gram staining, and a biochemical test [23].

#### 2.4.2. For Urine Samples

The urine samples were cultured into cysteine lactose electrolyte deficient (CLED) agar by semi-quantitative culture techniques using a standard calibrated loop (4 mm). A loop full of urine was streaked on the plate and then incubated at 37 °C in aerobic conditions for 18–24 h. Colony count was performed, in which a positive result was considered for plates having more than or equal to 10^5^ colony-forming units (CFU)/mL of urine based on Kass, Marple, and Sanford criteria [21,24]. 

#### 2.4.3. For Sputum, Wound Swab, Pus, Pericardial Fluids, Other Body Fluids, and Valve Tissues

These specimens were inoculated in BA and MA plates. The plates were incubated aerobically at 37 °C overnight. If growth occurred, then the isolated colony was identified by colony morphology, Gram staining, and a biochemical test [21,24]. 

### 2.5. Identification of the Isolates 

After incubation, the culture plates were examined for the growth of organisms. Presumptive identification of the isolates was made solely on the basis of colony morphology and Gram staining. Isolates were then subjected to other confirmatory biochemical tests (catalase, oxidase, sulphur indole motility, methyl red/voges-proskauer, triple sugar iron, citrate, urease, and oxidative fermentative test) [24].

### 2.6. Antibiotic Susceptibility Test of Isolated Organisms 

An antibiotic susceptibility test (AST) of isolated organisms was performed according to the CLSI guidelines [25]. The antibiotics used were ampicillin (10 µg), amikacin (30 µg), cotrimoxazole (25 µg), cephalosporin (5µg), nitrofurantoin (300 µg), nalidixic acid (10 µg), norfloxacin (10 µg), gentamicin (30 µg), ceftazidime (30 µg), cefotaxime (30 µg), cefepime (30 µg), imipenem (10 µg), meropenem (10 µg), piperacillin (100 µg), and piperacillin-tazobactam (100/10 µg). The susceptibility test was performed in vitro by the modified Kirby–Bauer disk diffusion method. In this method, the broth culture of the test organism (comparable to McFarland tube no. 0.5; inoculum density 1.5 × 10^8^ organisms/mL) was uniformly carpeted on the surface of Mueller Hinton agar (MHA). Appropriate antibiotic disks were placed onto the medium and then incubated at 37 °C for 18 h. After incubation, the inhibition zone was measured with the help of a measuring scale in mm and interpreted as susceptible, intermediate, or resistant by comparing the zones with standard interpretive criteria following CLSI 2019 guidelines [25]. 

### 2.7. Screening of Multidrug Resistant (MDR) and Potential ESBL Producers 

Isolates showing resistance to at least one agent of three or more classes of antimicrobial agents were classified as multi-drug resistant (MDR) [26]. If the zone of inhibition was ≤23 mm ceftriaxone, ≤21 mm ceftazidime, and ≤26 mm cefotaxime, the isolates were considered as a potential ESBL producer [25], and thus further subjected to the confirmatory tests.

### 2.8. Phenotypic Confirmation of ESBL Production

A combination disk test (CDT), as recommended by the CLSI (2018), was performed as the confirmation test for ESBL producers [27]. In this test, ceftazidime (30 µg) and cefotaxime (30 µg) disks alone and in combination with clavulanic acid (ceftazidime plus clavulanic acid, 30/10 µg; cefotaxime plus clavulanic acid, 30/10 µg) disks were applied onto an MHA plate (with an already inoculated test strain), and then incubated at 16–18 h for 35 ± 2 °C. Isolates showing an increase of ≥5 mm in the zone of inhibition of the combination disks in comparison to that of the ceftazidime/cefotaxime disk alone were confirmed as ESBL producers [27]. 

### 2.9. Preservation of the Isolates 

An axenic culture of ESBL producer *E. coli* isolates were preserved in 20% glycerol containing tryptic soya broth and kept at −70 °C until further processing for molecular assay [28,29].

### 2.10. Plasmid DNA Extraction and Amplification

All of the phenotypically confirmed ESBL producers *E. coli* were analyzed for detection of β-lactamase genes (*bla*_TEM_ and *bla*_CTX-M_). The test organisms were inoculated in Luria-Bertani (LB) broth and incubated overnight at 37 °C with aeration by using a water bath shaker. After incubation, the plasmid DNA was extracted by using the alkaline lysis method [30]. After extraction of the DNA, the DNA was suspended in 50 µL of TE buffer and stored at deep freeze (4 °C) or at −20 °C [30].

### 2.11. DNA Amplification and Detection

Conventional PCR was used to detect the plasmid genes. PCR amplification reactions were carried out in a 21 µL volume, in which a master mix containing 200 µM of dNTP_s_ (dATP, dCTP, dGTP, and dTTP), 120 nM of each primer (forward and reverse), 0.5 U/µL of *Taq polymerase* in 1× PCR buffer, 25 mM of MgCl_2_, and 3 µL of DNA template was added. Amplification reactions were performed in a DNA thermal cycler under the following thermal and cycling conditions for the *bla*_TEM_ (F.P:5′-GAGACAATAACCCTGGTAAAT-3′R.P:5′-AGAAGTAAGTTGGCAGCAGTG-3′) [16] and *bla*_CTX-M_ (Forward Primer: 5′-TTTGCGATGTGCAGTACCAGTAA-3′, Reverse Primer: 5′-CGATATCGTTGGTGGTGCCATA-3′) [31] genes: initial denaturation at 94 °C for 1 min, denaturation at 95 °C for 5 min of 35 cycles, annealing at 55 °C for 1 min of 35 cycles for *bla*_CTX-M_ and 56 °C for 45 s of 35 cycles for *bla*_TEM_, extension at 72 °C for 1 min of 35 cycles, and final extension at 72 °C for 10 min [16,31].

After PCR amplification, 15 µL of each reaction were separated by electrophoresis in 1.5% agarose gel for 60 min at 100 V in 0.5× TBE buffer. Ten µL of DNA was stained with ethidium bromide (5 µg/mL). Thus, amplified DNA bands were visualized by using a UV transilluminator [28,29]. The amplicon size for *bla*_TEM_ was 459 bp [16] and *bla*_CTX-M_ was 544 bp [31].

### 2.12. Quality Control

For the standardization of the drug susceptibility test, control strains of *K. pneumoniae* (ATCC 700603) and *E. coli* (ATCC 25922) were used. For, PCR control sterile water (negative) and the known positive DNA and negative controls from the previous extraction (positive) were processed to ensure the correctness of PCR process.

### 2.13. Statistical Analysis

Collected data were entered into SPSS version 24.0 software and analyzed. The chi-squared test was used to explore the association between categorical variables. A *p*-value of < 0.05 was considered to be significant in determining the relationship between the variables.

## 3. Results

### 3.1. Distribution of Culture-Positive Bacterial Isolates

Out of the total samples, 17.8% (*n* = 190/1065) showed bacterial growth. Among 190 bacterial isolates, 57.4% (109/190) were Gram-negative bacteria, and 42.6% (81/190) were Gram-positive bacteria. Among 109 Gram-negative bacteria, 40.3% (44/109) were *E. coli*, 30.2% (33/109) were *K. pneumoniae*, and 11.0% (12/109) were *A. baumannii*.

Among 190 patients, 46.3% (88/190) were male. Among the age-wise distribution of patients, 42.6% (81/190) were from the age group >46 years, followed by the age group 15–45 years (41.1%; 78/190) (Table 1).

### 3.2. Antibiotic Susceptibility Pattern of Isolated Gram-Negative Bacteria

In this study, carbapenem drugs, like meropenem and imipenem, were found to be more effective than other antibiotics. Among 44 isolates of *E. coli*, 91.0% (40/44) were sensitive to meropenem, followed by imipenem (86.4%; 38/44), gentamicin (72.8%; 38/44), and amikacin (66%; 29/44), whereas the majority (75.0%; 33/44) of isolates were resistant to ampicillin, cotrimoxazole (63.6%; 28/44), norfloxacin (63.6%; 28/44), and nalidixic acid (61.3%; 27/44). The antibiotic susceptibility profile of other Gram-negative isolates is presented in Table 2.

### 3.3. Multidrug Resistance (MDR) among Gram-Negative Organisms

Among 109 Gram-negative bacterial isolates, more than half (51.3%; 56/109) were MDR. Among 56 MDR cases, the highest MDR was detected in the age group >45 years (50.9%; 27/56), followed by 16–45 years (35.8%; 19/56). MDR cases were found more in males (58.5%; 31/56) compared to females (41.5%; 22/56) (*p* = 0.05). In the specimen-wise distribution of MDR cases, the highest MDR cases were detected in urine specimens (45.3%; 24/56), followed by pus/wound swab (28.3%; 15/56) and sputum (20.7%; 11/56). There was a significant association between the clinical specimens and MDR bacteria (*p* = 0.02) (Table 3).

The highest percentage of MDR was seen in *Escherichia coli* (63.6%; 28/44), followed by *Acinetobacter baumannii* (58.3%; 7/12), *Klebsiella pneumoniae* (54.5%; 18/33), *Citrobacter freundii* (50.0%; 1/2), and *Pseudomonas aeruginosa* (22.2%; 2/9). *Proteus mirabilis* and *Serratia marcescens* were not found as MDR (Figure 1).

### 3.4. Distribution of ESBL Producers ESBL-Producing E. coli

Among 44 *E. coli* isolates, 22 isolates were screened positive for ESBL production using ceftazidime, whereas 24 isolates were screened positive for ESBL production using cefotaxime. In the confirmatory assay, 27.3% (12/44) were confirmed as ESBL-producing *E. coli*. Among 12 ESBL producers, the majority (66.7%; 8/12) were from females. All of the ESBL producers (*n* = 12) were detected in urine samples. Statistically, there was no significant association between the type of specimen and ESBL producer (*p* = 0.54) (Table 4).

### 3.5. Antibiotic Susceptibility Pattern of ESBL-Producing Escherichia coli

All 12 ESBL-producing *Escherichia coli* isolates were found to be 100% resistant against ampicillin, cefotaxime, and ceftazidime. Most of the ESBL producers were found to be susceptible to meropenem (91.6%; 11/12), imipenem (83.4%; 10/12), gentamicin (*n* = 7; 58.3%; 7/12), and amikacin (*n* = 6; 50%; 6/12) (Figure 2).

### 3.6. Molecular Detection of ESBL Producer Genes

Among 12 ESBL-producing *E. coli*, 58.4% (*n* = 7), 41.6% (*n* = 5), and 25.0% (*n* = 3) isolates were tested positive for the *bla*_CTX-M_ gene, *bla*_TEM_ gene, and both *bla*_CTX-M_ and *bla*_TEM_ genes, respectively (Figure 3). The *bla*_CTX-M_ gene and *bla*_TEM_ gene on 1.5% agarose gel under UV light was detected with a band of product size of 544 bp and 499 bp, respectively (Figure 4).

## 4. Discussion

### 4.1. Overall Findings

Infections mediated by Gram-negative bacteria (GNB) constitute the major burden in low- and middle-income countries (LMICs) due to acquisition of various resistant genotypes. In particular, *E. coli*, *K. pneumoniae*, *E. cloacae*, and non-fermentative bacteria, such as *P. vulgaris*, *A. baumannii*, and *Salmonella* species, represent the major members of GNB that are associated with frequent and more severe forms of clinical manifestations, including urinary tract infection, bacteremia, and pneumonia [10,32]. More specifically, *E. coli* and *K. pneumoniae* represent the most predominant pathogens isolated from such infections [15,33]. Antibiotics are the choice of therapeutic options for the management of these infectious diseases. However, recent findings from Nepal on the AMR suggest the emerging resistance to virtually all classes of frontline drugs in use [5]. Acquisition and horizontal transfer of resistant genes from multiple sources of pathogenic microbes, the environment, and animals are suggested as the principal drivers of the uncontrolled spread of resistance [5,34]. Urinary tract infection (UTI) was the major clinical complaint among patients in this study. Overall, more than half of the isolates showed multidrug resistance (MDR) to the common antibiotics in use. Similarly, half of the *E. coli* isolates were ESBL producers, of which almost one-third of the isolates had ESBL genes *bla*_CTX-M_ and *bla*_TEM_. A high prevalence of MDR and acquisition of resistant genotypes is attributed to the poor infection control in the country, and warrants urgent efforts to tackle the burgeoning AMR.

In this study, urine constituted the highest frequency (28.2%) among all of the clinical samples. Our samples echo previous studies conducted in the International Friendship Hospital, Kathmandu [3], Universal College of Medical Sciences Bhairahawa [10], Human Organ Transplant Center [33], and Nobel Medical College, Biratnagar [35]. This may be due to the high association of Gram-negative bacteria with UTI, which is the most common among patients attending a hospital in Nepal [16]. Similarly, *E. coli* was the most common organism, followed by *Klebsiella* spp., *A. baumannii*, and *Pseudomonas* spp. *E. coli* alone constituted almost one-fifth of the total isolates and more than one-third of the GNB alone. Our findings are consistent with the previous studies from Everest Hospital, Kathmandu [20], Alka Hospital, Lalitpur [28], International Friendship Hospital, Kathmandu [3], Padma Nursing Hospital, Pokhara [36], National Public Health Laboratory, Kathmandu [37], B.P. Koirala Institute of Health Sciences, Dharan [38], and Alka Hospital, Lalitpur [39].

### 4.2. Antibiotic Resistance and Multidrug Resistance

In this study, most of the GNB showed resistance to the commonly prescribed broad-spectrum antibiotics. In addition to this, almost all of them were sensitive to the carbapenem drugs. More specifically, *E. coli* and *K. pnemoniae* were highly sensitive towards carbapenems (imipenem, meropenem) and amikacin. *E. coli* showed high sensitivity (>80%) towards meropenem and imipenem in our study. These findings are in line with the study reported from Manmohan Memorial Medical College and Teaching Hospital (MMCTH) [18], Alka Hospital, Lalitpur, Kathmandu [28]. The antibiogram of this study suggests the utility of carbapenems as a secondary therapeutic option for infections caused by MDR Gram-negative organisms. However, recent studies have shown the growing resistance to the last resort antimicrobials, carbapenems [40].

Overall, more than half (51.3%) of the GNB isolates were reported as MDR in our study. A high prevalence of MDR is reported from other studies from the International Friendship Hospital [3], Everest Hospital [20], Alka Hospital [28], Human Transplant Center, Kathmandu [33], Manmohan Memorial Medical College, and Teaching Hospital (MMCTH), Kathmandu [18]. The difference in the prevalence of MDR in Gram-negative bacteria in this study and previous studies is attributable to the differences in the source and types of samples, sample size, growth of organisms, and drug resistance pattern [10].

### 4.3. ESBL Producers and Acquisition of Resistant Genotypes

In this study, more than one-fourth (27.3%) of the total *E. coli* isolates were confirmed as ESBL producers. This finding is consistent with the previous studies reported from Model Hospital, Bagbazar, Kathmandu [15], Everest Hospital, Baneshwor, Kathmandu [20], and International Friendship Hospital, Maharajgunj, Kathmandu [3], and lower than a study reported from Manmohan Memorial Medical College and Teaching Hospital (MMCTH) [16,18] and Om Hospital and Research Centre, Kathmandu [41]. The higher rate of ESBL in this organism compared to the previous studies may indicate the temporal impact of increasing infections and AMR over the years. For instance, together with the growing population, the trend in self-medication, distribution of suboptimal quality of antibiotics, poor infection control, sanitation, and hygiene practices (in the personal and community level) are the driving mechanisms for the unabated spread of resistant pathogens [11,42].

In our study, all of the ESBL-producing *E. coli* (27.3%) harbored at least one of the tested genes (*bla*_CTX-M_ (58.4%) and *bla*_TEM_ (41.6%)), while one-fourth of them showed the co-expression of both genotypes. This result indicates a slight decrease in prevalence of *bla*_TEM_ and *bla*_CTX-M_ genes than a previous study reported from Annapurna Neurological Institute and Allied Sciences, Kathmandu (33.2% ESBL producers, 83.8% *bla*_TEM_, and 30.8% *bla*_CTX-M_) [31]. Another study reported 40.3% ESBL producers, 83.8% *bla*_TEM_ and 66.1% *bla*_CTX-M_ from MMCTH [16]. The dominance of the *bla*_CTX-M_ gene has been reported from previous studies from Nepal and overseas [43,44,45,46]. The difference in the prevalence rate of *TEM* and *CTX-M* genes between the present study and other studies might be due to the difference in the sample population and antibiotic susceptibility pattern of *E. coli*, the type of study design, and sample size. Similarly, co-expression (or multiple occurrences) of the genes were common in other studies too. These genes are often present in large plasmids, and are capable of conferring resistance to the organisms [47].

All of the ESBL-producing *E. coli* isolates were resistant towards ampicillin, cefotaxime, and ceftazidime, while the entire isolates were also susceptible to the carbapenems, such as meropenems and imipenems. Our findings are consistent with some previous findings reported from Manipal Teaching Hospital, Pokhara [48], International Friendship Hospital, Maharajgunj, Kathmandu [3], Chitwan Medical College, Bharatpur [49], International Friendship Children Hospital, Kathmandu [50], Kathmandu Medical College Teaching Hospital, Sinamangal, Kathmandu [14], and Shahid Gangalal National Heart Centre, Bansbari, Kathmandu [51,52]. Higher resistance to penicillin and third generation cephalosporin in this study can be solely attributable to their ability to produce ESBL.

A higher fraction (respectively, two-thirds and slightly less than two-thirds) of ESBL producers and MDR were isolated from the female patients. The slight preponderance in females is explained by the higher prevalence of UTIs among females. Similarly, women of the age group below adulthood and in the post-menopausal stage are infected due to hormonal changes and poor sanitation practices that are often prevalent in LMICs [36,37]. GNB are the potential agents for nosocomial infections, and the outcome of such infections are associated with prolonged hospital stays, increased ICU admissions, and unwanted morbidities and mortalities [3,53].

## 5. Strengths and Limitations

The prevalence of GNB, their antibiogram, and the status of MDR among clinical samples of the study hospital can serve as an important reference tool for future studies, clinicians, and policy makers. Exploration of the prevalence of resistant genotypes of ESBL-producing *E. coli* in this study highlights the importance and need of molecular diagnostic facilities for a more precise detection of the infectious diseases. However, our study also suffers from a number of limitations. Our study design was limited to a single hospital, and had small clinical samples, which cannot generalize the prevalence of AMR elsewhere. Moreover, due to the limitation of the laboratory resources and funding, we could not characterize the resistant genotypes of other Gram-negative organisms. In addition to this, we could not characterize the other members of the ESBL genes, including SHV within the same family, and the genotypes of β-lactamases of other Ambler classes. The study is limited by the inability to explore the origin of the resistant genotypes and their potential transferability. Therefore, future studies are recommended to include the larger populations and provide an extensive characterization of the resistant genotypes to yield an accurate burden of AMR.

## 6. Conclusions

More than half of the Gram-negative isolates were detected as the multidrug resistant strains in our study. Half of the pathogenic strains of *E. coli* were potential ESBL producers, of which half of them harbored the ESBL genes under question. The findings of this study imply an urgent need for early suspicion, identification, and AST to optimize treatment and limit the spread of AMR.

## Figures and Tables

**Figure 1 diseases-09-00015-f001:**
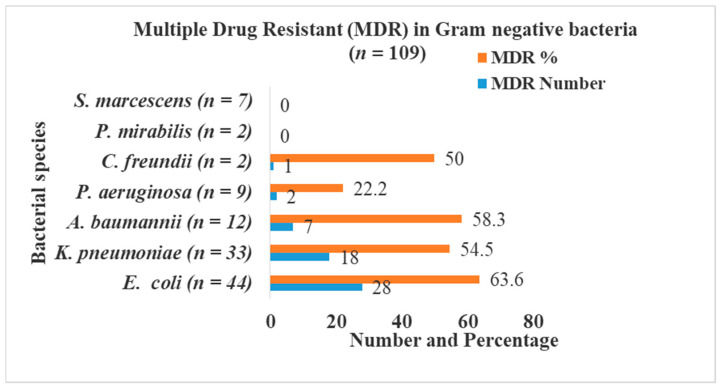
MDR in Gram-negative bacteria.

**Figure 2 diseases-09-00015-f002:**
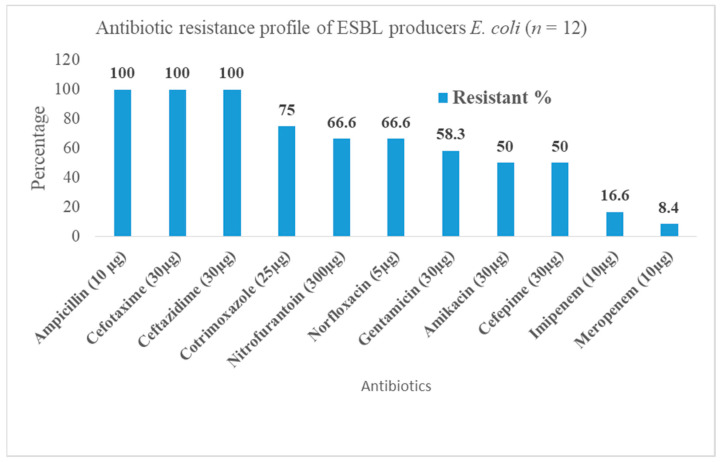
Antibiotic resistance profile of ESBL producers *E. coli*.

**Figure 3 diseases-09-00015-f003:**
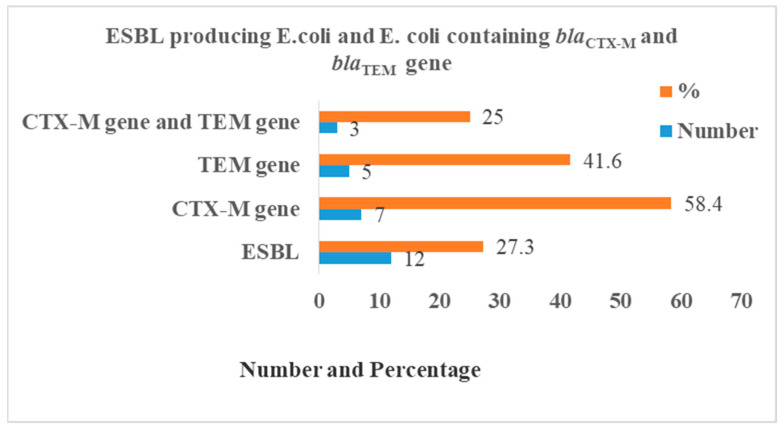
Distribution of *bla*_CTX-M_ and *bla*_TEM_ genes in *E. coli*.

**Figure 4 diseases-09-00015-f004:**
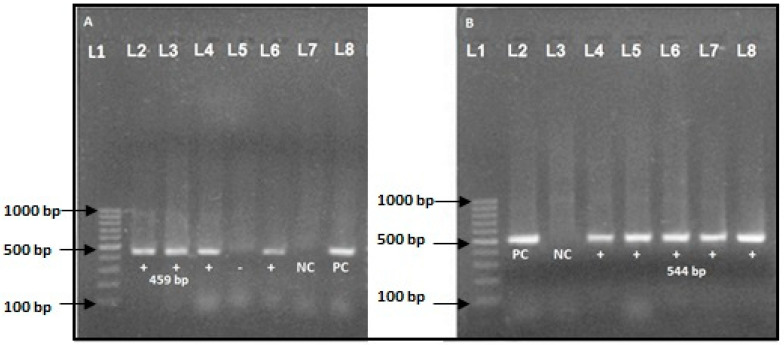
(**A**) *bla*_TEM_ gene (459 bp) (Lane L1, marker DNA (100–1000 bp)), L2, L3, L4, L6 positive band, *bla*_TEM_ gene (459-bp), L7: negative control, and L8: positive control. (**B**): *bla*_CTX-M_ gene (544 bp) (Lane L1, marker DNA (100–1000 bp)), L2: positive control, L3: negative control, L4, L5, L6, L7, and L8: *bla*_CTX-M_ gene (544 bp).

**Table 1 diseases-09-00015-t001:** Demographic characteristics and the distribution of culture-positive bacterial isolates (*n* = 190).

Character/Bacterial Isolates	Number	Culture Positive		*p*-Value
		Number	Percentage	
**Gender**				
Male	480	88	46.3	0.7
Female	585	102	53.7	
**Age groups (in years)**				
0–15	180	31	16.3	0.3
15–45	490	78	41.1	
>46	395	81	42.6	
**Type of specimens**				
Blood	246	13	6.8	1.8
Urine	304	92	48.4	
Sputum	280	29	15.3	
Pus/wound swab	151	40	21.1	
Catheter tips	59	10	5.3	
Body fluids	25	6	3.1	
**Type of bacteria**				
Gram-negative bacteria		109	57.4	
*E. coli*		44	40.4	
*Klebsiella pneumoniae*		33	30.3	
*Acinetobacter baumannii*		12	11.1	
*Pseudomonas aeruginosa*		9	8.3	
*Serratia marcescens*		7	6.3	
*Citrobacter* spp.		2	1.8	
*Proteus mirabilis*		2	1.8	
Gram-positive bacteria		81	42.6	
*Staphylococcus aureus*		39	48.2	
CONS		19	23.5	
*Enterococcus* spp.		15	18.5	
*Streptococcus* spp.		8	9.8	

**Table 2 diseases-09-00015-t002:** Antibiotic susceptibility pattern of Gram-negative bacterial isolates (*n* = 109).

Antibiotics	*E. coli* (*n* = 44)		*K. pneumoniae* (*n* = 33)		*A. baumannii* (*n* = 12)		*P. aeruginosa* (*n* = 9)		*S. marcescens* (*n* = 7)		*C. freundii* (*n* = 2)		*P. mirabilis* (*n* = 2)	
	Sensitive	Resistant	Sensitive	Resistant	Sensitive	Resistant	Sensitive	Resistant	Sensitive	Resistant	Sensitive	Resistant	Sensitive	Resistant
	N (%)	N (%)	N (%)	N (%)	N (%)	N (%)	N (%)	N (%)	N (%)	N (%)	N (%)	N (%)	N (%)	N (%)
Ampicillin (10 µg)	11 (25.0)	33 (75.0)	10 (30.4)	23 (69.6)					3 (42.9)	4 (57.1)	0	2 (100)	1 (50.0)	1 (50.0)
Amikacin (30 µg)	29 (66.0)	15 (34.0)	15 (45.5)	18 (54.5)	6 (50)	6 (50)	6 (66.7)	3 (33.3)	6 (85.8)	1 (14.2)	1 (50.0)	1 (50.0)	2 (100%)	0
Cotrimoxazole (25 µg)	16 (36.4)	28 (63.6)	13 (39.4)	20 (60.6)					5 (71.5)	2 (28.5)	1 (50.0)	1 (50.0)	2 (100%)	0
Nitrofurantoin (300 µg)	18 (41.0)	26 (59.0)	12 (36.4)	21 (63.6)					-	-	1 (50.0)	1 (50.0)	2 (100%)	0
Nalidixic acid (30 µg)	17 (38.7)	27 (61.3)	11 (33.4)	22 (66.6)					-	-	1 (50.0)	1 (50.0)	0	2 (100)
Norfloxacin (5 µg)	16 (36.4)	28 (63.6)	-	-					-	-	1 (50.0)	1 (50.0)	0	2 (100)
Gentamicin (30 µg)	32 (72.8)	12 (27.2)	17 (51.6)	16 (48.4)	6 (50)	6 (50)	7 (77.8)	2 (22.2)	7 (100)	0	2 (100)	0	2 (100)	0
Ceftazidime (30 µg)	22 (50.0)	22 (50.0)	21 (63.7)	12 (36.3)	7 (58.4)	5 (41.6)	7 (77.8)	2 (22.2)	7 (100)	0	0	2 (100)	2 (100%)	0
Cefotaxime (30 µg)	20 (45.5)	24 (54.5)	22 (66.7)	11 (33.3)					7 (100)	0	0	2 (100)	2 (100)	0
Cefepime (30 µg)	24 (54.6)	20 (45.4)	23 (69.7	10 (30.3)	7 (58.4)	5 (41.6)	6 (66.7)	3 (33.3)	7 (100)	0	0	2 (100)	2 (100%)	0
Imipenem (10 µg)	38 (86.4)	6 (13.6)	29 (87.9)	4 (12.1)	8 (66.7)	4 (33.3)	9 (100)	0 (0)	-	-	2 (100)	0	-	-
Meropenem (10 µg)	40 (91.0)	4 (9.0)	30 (91.0)	3 (9.0)	9 (75.0)	3 (25.0)	9 (100)	0 (0)	-	-	2 (100)	0	-	-
Piperacillin (100 µg)					4 (33.4)	8 (66.6)	6 (66.7)	3 (33.3)						
Piperacillin-tazobactam (100 µg/10 µg)					5 (41.7)	7 (58.3)	8 (89.0)	1 (11.0)						
Ciprofloxacin (5 µg)							4 (44.5)	5 (55.5)						

**Table 3 diseases-09-00015-t003:** Prevalence of MDR in Gram-negative bacteria according to demographic characteristics (*n* = 109).

Character	MDR		Non MDR		*p*-Value
	Number	%	Number	%	
**Age Group (in years)**					
0–15	9	16.1	7	13.2	0.5
16–45	24	42.9	19	35.8	
>45	23	41.1	27	50.9	
**Gender**					
Male	22	39.3	31	58.5	0.05
Female	34	60.7	22	41.5	
**Types of specimens**					
Blood	7	12.5	1	1.9	0.02
Urine	38	67.8	24	45.3	
Sputum	3	5.4	11	20.7	
Pus/wound swab	5	8.9	15	28.3	
Catheter tips	3	5.4	2	3.8	
**Type of bacteria**					
*E. coli*	28	50	16	30.3	0.01
*Klebsiella pneumoniae*	18	32.1	15	28.3	
*A. baumannii*	7	12.4	5	9.4	
*Pseudomonas aeruginosa*	2	3.6	7	13.2	
*Citrobacter* spp.	1	1.9	1	1.8	
*Serratia marcescens*	0	0	7	13.2	
*Proteus mirabilis*	0	0	2	3.8	

**Table 4 diseases-09-00015-t004:** Prevalence of ESBL producers, *bla*_CTX-M_, and *bla*_TEM_ genes in *E. coli* according to age, sex, and types of specimens.

Character	ESBL Producer (*n* = 12)	*bla*_CTX-M_ Gene (*n* = 7)	*bla*_TEM_ Gene (*n* = 5)
Number	%	*p*-Value	Number	%	*p*-Value	Number	%	*p*-Value
**Age group (in years)**									
0–15	1	8.3	0.33	1	14.3	0.65	0		0.65
16–45	7	58.3		4	57.1		3	60	
>45	4	33.3		2	28.6		2	40	
**Gender**									
Male	4	33.3	0.53	4	57.1	0.08	1	20	0.57
Female	8	66.7		3	42.9		4	80	
**Types of specimens**									
Blood	0	0	0.54						
Urine	12	100		7	100		5	100	
Sputum	0	0							
Pus/wound swab	0	0							
Catheter tips	0	0

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
