# Peer review of "Detection of TEM and CTX-M Genes in Escherichia coli Isolated from Clinical Specimens at Tertiary Care Heart Hospital, Kathmandu, Nepal"

_diseases, 2021, doi:10.3390/diseases9010015_

Round 1

Reviewer 1 Report

This study presents the AMR of GN from patients in a select hospital in Nepal. The findings have some clinical significance and the ms is overall well-written. My comments are as follows:

  1. Writing, please check the gene name throughout the ms. First of all, all gene names should be italics and the ESBL genes should be writing in their standard styles. E. g. blaTEM but not TEM or blaTEM. In addition, there should be a space between the value and the unit. E.g. 25 ml but not 25ml. The other inappropriate writing: e.g. line 175, MgCl2 but not MgCl2; In table 2, first row, “P. mirabilis (n=2” you lost another half of bracket; line 137, in vitro should be italics; etc.

  1. My great concern is a about the bacterial determination. According to you statement in Abstract (Lines 23-24) and results section 3.1, you isolated 190 bacterial strains from 1065 samples, and you determined 44 E. coli, 33 Kp, 12 Ab, etc. My questions are: (1) The other 875 (1065-190) samples do not show any bacterial growth, right? I personally think this result is unbelievable for some samples, especially you used both MacConkey agar and blood agar. So do you compare the results of bacterial isolation to human symptoms? (2) For the plates with bacterial growth, I think there should be multiple bacterial colonies on one plate, but according to you results, you only got one E.coli colony from one plate (also the same for the other bacterial species). How could you make sure that there is only one E.coli colony on the plate? Also, how could you make sure all E. coli isolates from the same plate are the same? (I mean they might belong to different serotypes, or genotypes. For the other bacterial species are the same). (3) You have only mentioned seven GN bacterial species (Table 1). Did you examine all the bacterial colonies on each of the plates? If not, how could make sure there is no colonies of the other bacterial species being growth? I think all of these questions should be addressed clearly in the ms.

  1. Indeed you have a statement of ethical approval at the end of this ms, but this approval is only from the University. Since your study involved in the use of human samples from a hospital, an ethic approval from the hospital is also necessary.

  1. Table 2, first column, for the unites below each of the antibiotics, what do you mean “10 µg”, “25 µg”, …, please state clearly.

  1. References for some places are required. E.g. Line 168 a reference should be following alkaline-lysis method; line 274 a reference should be following in use (recent findings, from where?).

  1. Line 273 antimicrobial resistance (AMR), you do need to repeat it here since you have shown AMR is for antimicrobial resistance at the beginning of the ms. Just use AMR here is OK.

Reviewer 2 Report

Epidemiological studies of antibiotic-resistant bacteria is an important and hot topic. This manuscript is in the line of such analyses. The authors have collected quite a good clinical material which could be used for interesting analyses. Unfortunately, the limitations - which are actually well recognized by the authors themselves and listed in Section 4 - are significant enough to consider this study as a preliminary report. Particularly, low number of ESBL-producing strains do not allow to make solid conclusions on the basis of statistical analyses. Moreover, other bla-type genes, like bla(SHV) should be analyzed. Importantly, analyses of other bacteria found in this study should be analyzed, but the authors declared that their resources are too limited to perform such studies. On one hand, I appreciate the hard work of authors which was conducted under difficult economical conditions. On the other hand, without experiments mentioned above, the current state of the work must be considered as preliminary study.This is the main reason for my reservation.

Some other, specific points are listed below:

  1. All abbreviations used in this manuscript should be explained when they are mentioned for the first time in the text. This concerns also Abstract.
  2. Conclusions in the Abstract are trivial, particularly the last sentence is obvious in the light of a large numer of previously published reports in this field.
  3. Line 35: Please, specify what kind of infections is mentioned. E. coli and Klebsiella are not causative pathoges in all infections, i.e. infections of all tissues and organs.
  4. Section 2.6. It should be mentioned that amounts of antibiotics are per disk.
  5. Section 2.11. Times of particular steps of PCR are very long, especially denaturation (15 min !!!!). This is very unusual. Please, check the procedures if appropriate parameters were included into the description of this method.
  6. Section 2.13. It should be indicated what statistical test were used in particular analyses performed in this study.
  7. Figure 2. It is not recommended to show values of both resistant and sensitive strains. One of these values is sufficient.
  8. English usage is understandable, but some errors should be corrected.

Reviewer 3 Report

The following submitted manuscript by Sah et al. chronicles the prevalence of antibiotic resistance of bacteria taken from patients of a single Nepalese hospital suffering from heart infection. The study analyzes 1065 specimens taken from various sample sources (urine, blood, etc.) and characterizes bacterial content and antibiotic resistance (including multidrug resistance), along with correlations between type of bacterial infection and various age and gender demographics.  Motivation for this study stems from the rapid emergence of antimicrobial resistant pathogens caused by the overuse of such drugs in humans, animals and the environment.  Detailed analysis of the most common bacteria found associated with the infections, along with prevalence of antibiotic resistance, can help serve as a reference tool for healthcare workers and policy makers to establish guidelines for antibiotic use that most efficiently reduces the spread of antimicrobial resistance while providing effective therapies for patients.  The authors found that ~18% of the samples taken had culturable bacteria (using established protocols) with slightly more than half being classified as gram-negative, and E. coli being the predominant species followed closely by K. pneumoniae.  A large portion of the E. coli isolates were screened positive for ESBL production and all showed resistance against commonly used antibiotics such as ampicillin and cefotaxime. Importantly, the most efficient antibiotic against the identified gram-negative bacteria was the carbapenem drugs meropenem and imipenem. The manuscript contains additional important and relevant data as well.

Overall, the manuscript is concise and well written, and the data figures are clear and useful.  Apart from a few minor comments (see below) I would recommend the manuscript for publication.

Minor comments

Lines 34-35:  change “….causative pathogens in most of the infections.” To “….causative pathogens in most infections.”

Lines 47-49: The following sentence is a little unclear. I would reword the sentence to the following “…Gram negative bacteria have developed resistance to one of the most effective drugs (B-lactams) by producing enzymes with the ability to hydrolyzes bonds constituting the beta-lactam rings of the antibiotics.”

Lines 51 and 53: add relevant citation at the end of sentence

Line 56: delete “(microorganisms)”

Line 68: reword sentence from “…facilities can result the improper…” to “…facilities can result in the improper…”

Lines 74 and 75: add relevant citation at the end of sentence

Line 200: delete the word “the” from the sentence

Round 2

Reviewer 1 Report

The authors have almost addressed my concerns in the revised version, and I do not have any other comments at this time.

Reviewer 2 Report

The authors have addressed only minor specific points, but did not answer the major questions which caused my serious reservations to this manuscript. The authors did not even comment my major critical points. The subject of this study is important, but I still think this report is highly preliminary and not suitable for publication in this journal in its current form, particuarly without additional experiments.